# Reactive Epidermal Hyperplasia and Angiogenesis of the Rear (REAR): A Proposed Unifying Name for Senile Gluteal Dermatosis and Prurigiform Angiomatosis

Michelle Weiting Liang *, Joel H. L. Lim, Hui Yi Chia, Shang-Ian Tee and Joyce S. S. Lee

National Skin Centre, Singapore 637551, Singapore
* Correspondence: mliang@nsc.com.sg

**Abstract:** Senile gluteal dermatosis (SGD) is a common but seldom recognized condition. It is characterized clinically by unilateral or bilateral hyperkeratotic, lichenified plaques on the gluteal area, being attributed to prolonged sitting, particularly in the elderly. SGD also encompasses the recently proposed entity of prurigiform angiomatosis. Histologically, there are features of lichenification, such as epidermal hyperplasia and a preserved granular layer, with prominent dermal angioproliferation. We report 4 cases of this condition as well as novel findings of variably increased mast cells and superficial lymphatic vessels in addition to the proliferation of dermal blood vessels. We propose a unifying name for Reactive Epidermal hyperplasia and Angiogenesis of the Rear (REAR) to encapsulate the characteristic clinical and histological features of this distinct entity.

**Keywords:** senile gluteal dermatosis; pressure dermatitis; prurigiform angiomatosis; sitter's triangle





## 1. Introduction

Senile gluteal dermatosis (SGD) is a common but seldom recognized condition, mainly reported in elderly patients in association with prolonged sitting. It is characterized by unilateral or bilateral hyperkeratotic, lichenified plaques on the gluteal area. Histological features reported have been described as mild and nonspecific, with hyperkeratosis, psoriasiform epidermal hyperplasia, vascular dilatation/proliferation in the upper dermis, and reactive lymphohistiocytic perivascular infiltrate. The clinical and histological features of SGD encompass the entity proposed as prurigiform angiomatosis, which is characterized by the clinical presentation of prurigo/lichen simplex chronicus-like epidermal hyperplasia with prominent dermal angioproliferation. We report four cases of this condition with the aim of highlighting a distinct triad of clinical and histopathological findings of this entity, namely (1) the sitter's triangle formed by the coccygeal apex and the ischial tuberosities in a person with a history of prolonged sitting, in association with (2) histopathological features of lichenification, and (3) prominent reactive vascular proliferation within the dermis. The authors also propose a unifying name for Reactive Epidermal hyperplasia and Angiogenesis of the Rear (REAR), which highlights both the characteristic clinical and histological features of this condition.

## 2. Case Descriptions

The first patient is a 68-year-old Chinese male with hypertension and gastrointestinal reflux disease. He presented with rashes over both buttocks for 2 years, with intermittent pain on sitting. The rash did not improve with topical mometasone ointment. On examination, there were symmetrical erythematous plaques on bilateral buttocks (Figure 1a), inferiolateral to the coccygeal apex (Figure 1b), with some superficial scaling and surrounding mild lichenification exhibiting horizontal hyperkeratotic ridges. Skin biopsy from the left buttock plaque showed hyperkeratosis with psoriasiform hyperplasia (Figure 2a,b). Within the upper to mid-dermis was a proliferation of capillary-type vessels without any overt

endothelial cell atypia (Figure 2c). There was also a modest increase in the number of lymphatic vessels interspersed between blood vessels on the podoplanin stain (Figure 2d). Interestingly, mast cell (MC) density was increased, as highlighted on the chloroacetate esterase (CAE) stain (Figure 2e,f).

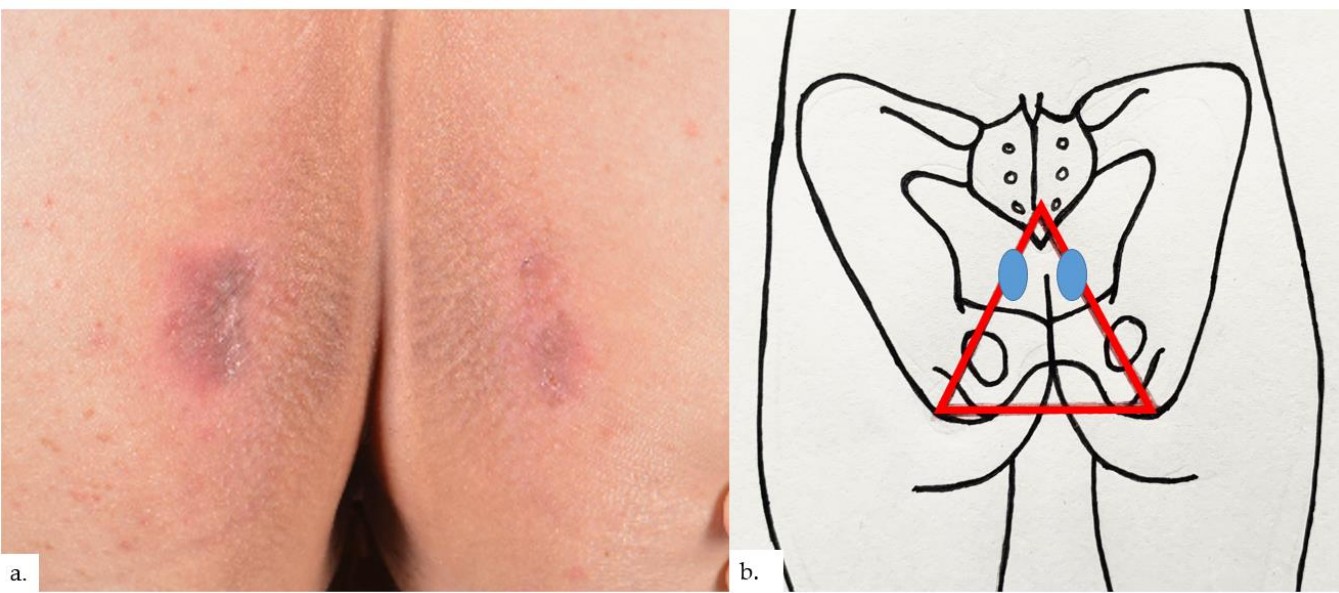

**Figure 1.** Patient 1. (**a**) Presence of symmetrical erythematous plaques over both buttocks, with some superficial scaling and surrounding lichenification showing characteristic horizontal hyperkeratotic ridges; (**b**) the plaques (blue ovals) were inferio-lateral to the coccygeal apex, at the apex of the sitter's triangle (red triangle).

The second patient is a 73-year-old Chinese male with diabetes mellitus, hypertension, hyperlipidemia, asthma, benign prostatic hyperplasia, cognitive decline, and pulmonary tuberculosis. He presented with a lump over his right buttock of 2 years duration, which had been growing recently, with some mild discomfort on sitting down. On examination, there were erythematous plaques on the bilateral buttocks, with scaling and lichenification (Figure 3a). The plaques were slightly asymmetrical, with the one on the left just lateral to the coccygeal apex, whilst the one on the right was located more inferior-medially at the intergluteal fold (Figure 3b), both within the upper portion of the sitter's triangle. Notably, there was blanchable erythema over the natal cleft. Skin biopsy from the right buttock plaque showed compact hyperkeratosis and psoriasiform hyperplasia of the epidermis with hypergranulosis, features consistent with lichenification (Figure 4a). There was proliferation and dilatation of thin-walled vessels within the upper to the mid dermis (Figure 4b,c). Notably, most of the vascular proliferation within the upper dermis were the lymphatics, as demonstrated on podoplanin (Figure 4d). The elevated number of MC was delineated on CAE in association with the increased vasculature (Figure 4e,f).

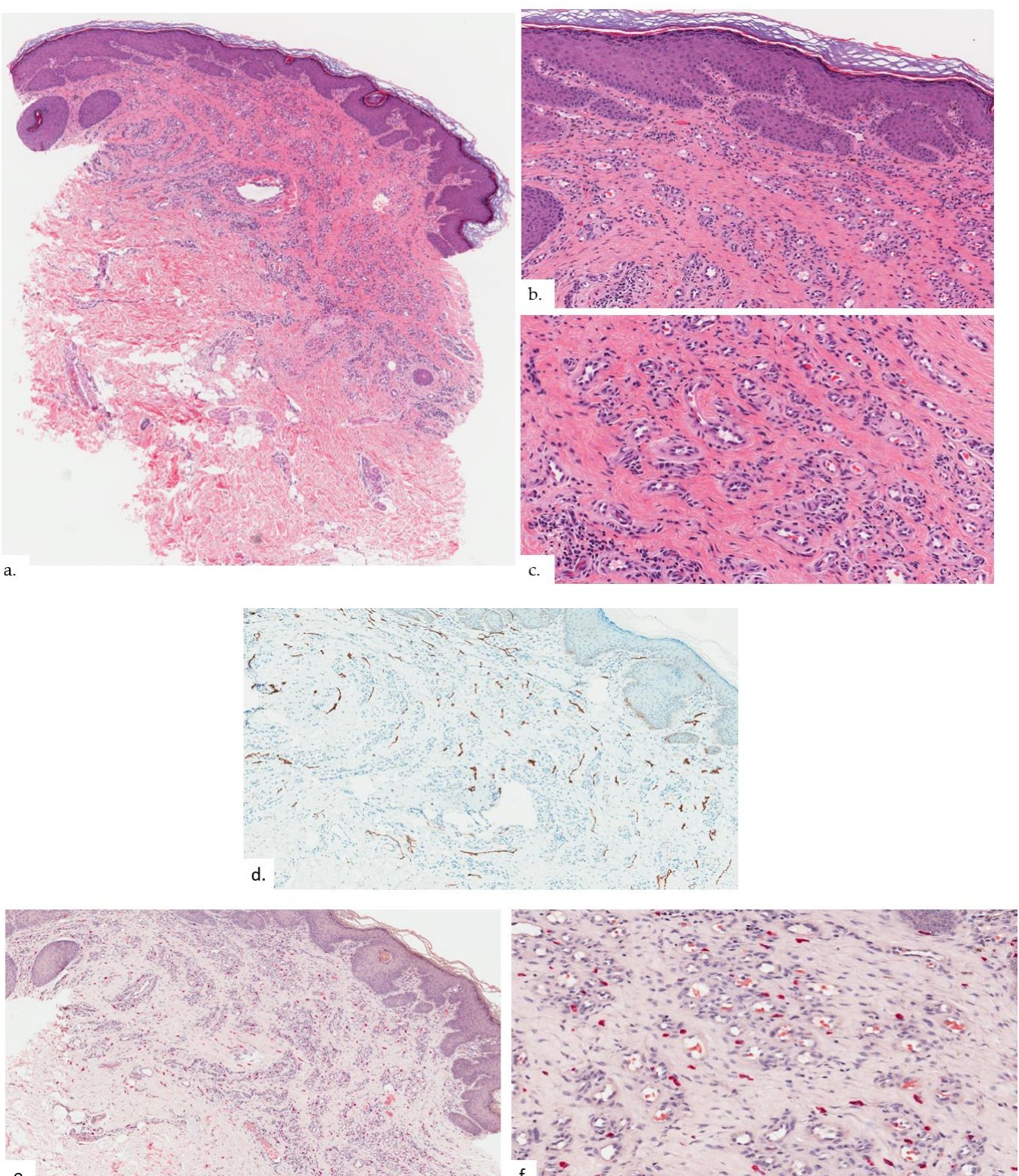

**Figure 2.** Patient 1. (**a**,**b**) There was hyperkeratosis with psoriasiform hyperplasia; (**c**) there was prominent vascular proliferation present in the upper to the mid dermis, consisting of proliferation of capillary type vessels without any overt endothelial cell atypia; (**d**) a modest increase in lymphatic channels was highlighted on podoplanin; (**e**,**f**) MC density was increased in proportion with vasoproliferation as seen on CAE stain.

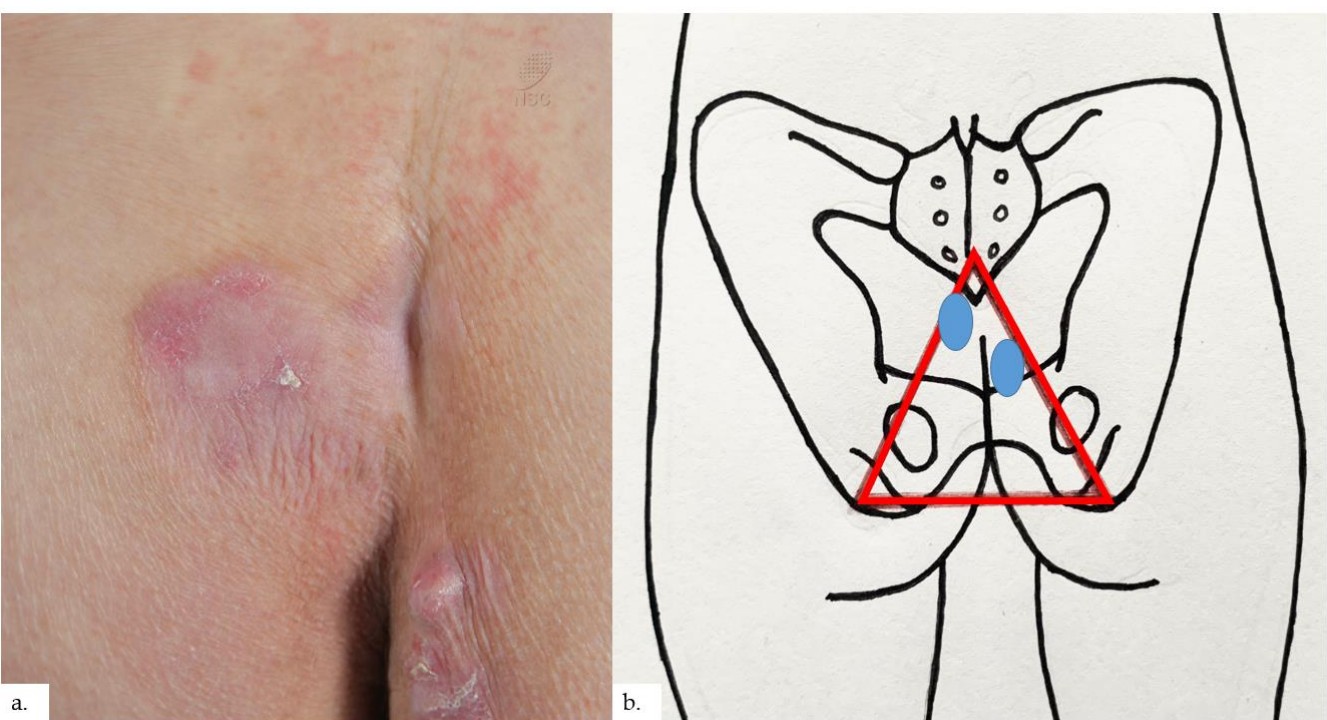

**Figure 3.** Patient 2. (**a**) There were bilateral erythematous plaques over his buttocks, with scaling and lichenification; notably, there was blancheable erythema over the natal cleft; (**b**) the plaques (blue ovals) were slightly asymmetrical, with the one on the left just lateral to the coccygeal apex, whilst the one on the right was located more inferior-medially at the intergluteal fold, within the upper portion of the sitter's triangle (red triangle).

The third patient is a 68-year-old Chinese male with a history of foot eczema. He presented with a right buttock lump of 3 months duration. There was some discomfort in sitting. There had been some improvement with previous cryotherapy and the use of beprosalic ointment. On examination, there was a unilateral erythematous plaque over the right intergluteal fold, just inferio-lateral to the coccygeal apex, with scaling and surrounding erythema (Figure 5a,b). Skin biopsy from the plaque showed compact hyperkeratosis and psoriasiform hyperplasia with hypergranulosis (Figure 6a). There was notable vascular proliferation within the upper dermis (Figure 6b), consisting of a proliferation of capillary-type vessels (Figure 6c) with a modest increase in lymphatic channels (Figure 6d) as well as increased MC quantity (Figure 6e,f).

The final patient is a 94-year-old Chinese male with a history of hypothyroidism, atrial fibrillation, and benign prostatic hyperplasia. He presented with a lump over the left buttock for a few months, which had become painful in the last few weeks. On examination, there was a small tender 0.5 cm rounded dermal centered nodule over the left gluteal fold with hyperpigmentation and lichenification of the surrounding skin (Figure 7a). The nodule was overlying the ischial tuberosity at the base of the sitter's triangle (Figure 7b). An excision biopsy was performed. Histopathological examination showed compact hyperkeratosis and hyperplasia of the epidermis and follicular epithelium, with vascular proliferation throughout the dermis extending into the subcutis (Figure 8a). The epidermis was hyperplastic, with follicular plugging (Figure 8b). A focal foreign body granuloma was seen, with multinucleated giant cells surrounding keratinaceous debris (Figure 8c). Infective stains were negative. There was an exuberant proliferation of capillary-type vessels throughout the dermis, extending into the subcutis (Figure 8d), with an increase in lymphatic channels on the podoplanin stain (Figure 8e), as well as a proportionate increase in MC density (Figure 8f,g).



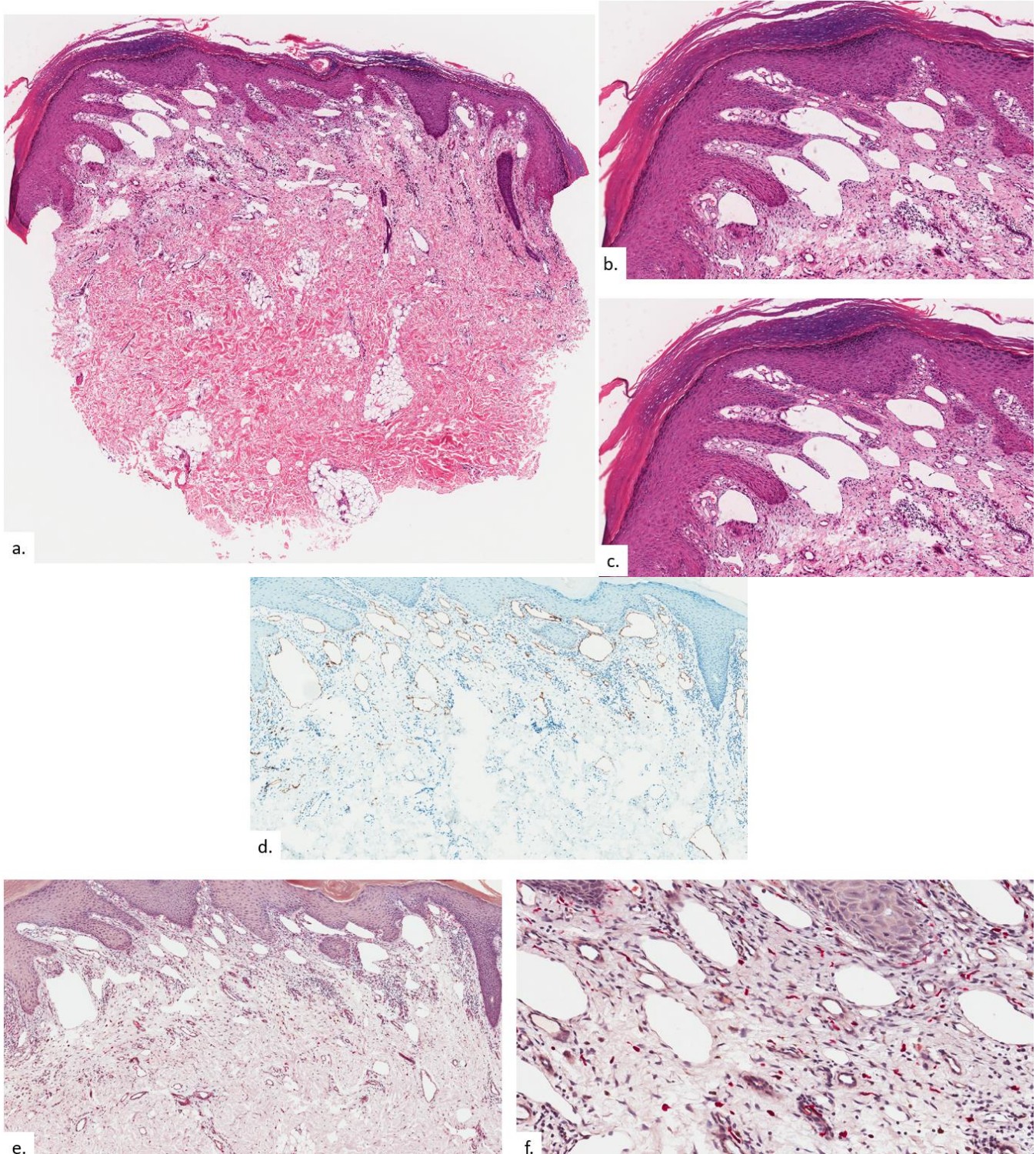

**Figure 4.** Patient 2. (**a**,**b**) There is compact hyperkeratosis and psoriasiform hyperplasia of the epidermis with hypergranulosis; (**c**) there was a vascular proliferation within the upper to mid dermis, consisting of a greater proportion of lymphatic channels than blood vessels, as seen on podoplanin (**d**); (**e**,**f**) MC numbers were significantly elevated as highlighted on CAE.

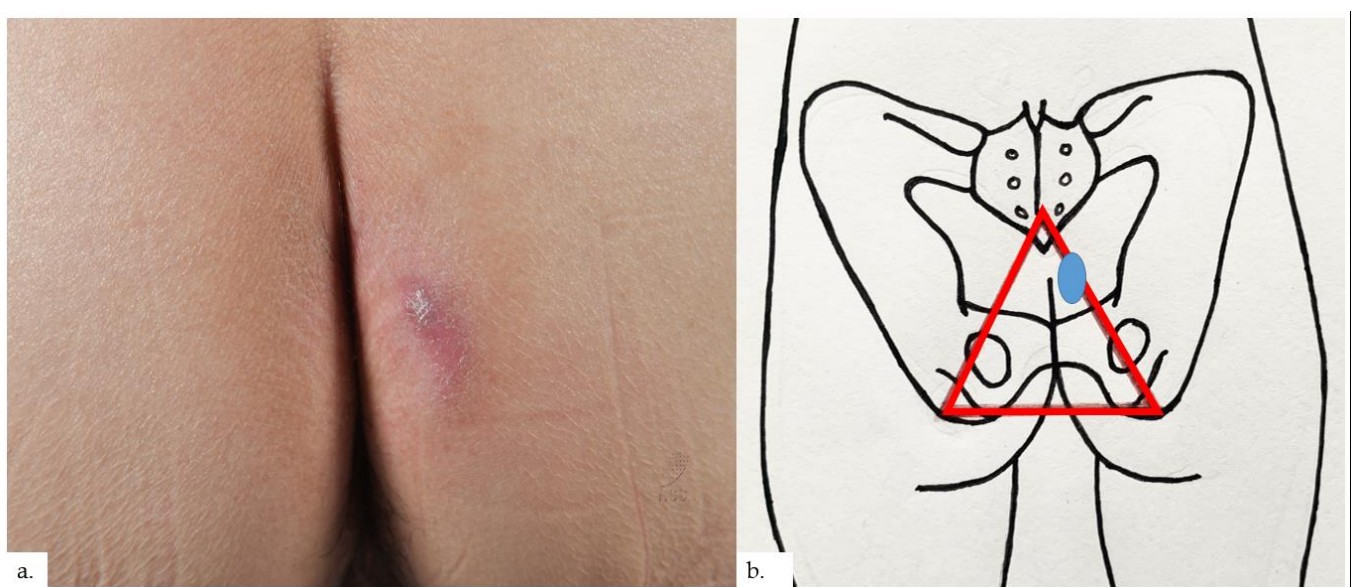

**Figure 5.** Patient 3. (**a**) There was a unilateral erythematous plaque over the right buttock with scaling and surrounding erythema; (**b**) the plaque (blue oval) was at the right intergluteal fold, within the sitter's triangle (red triangle).

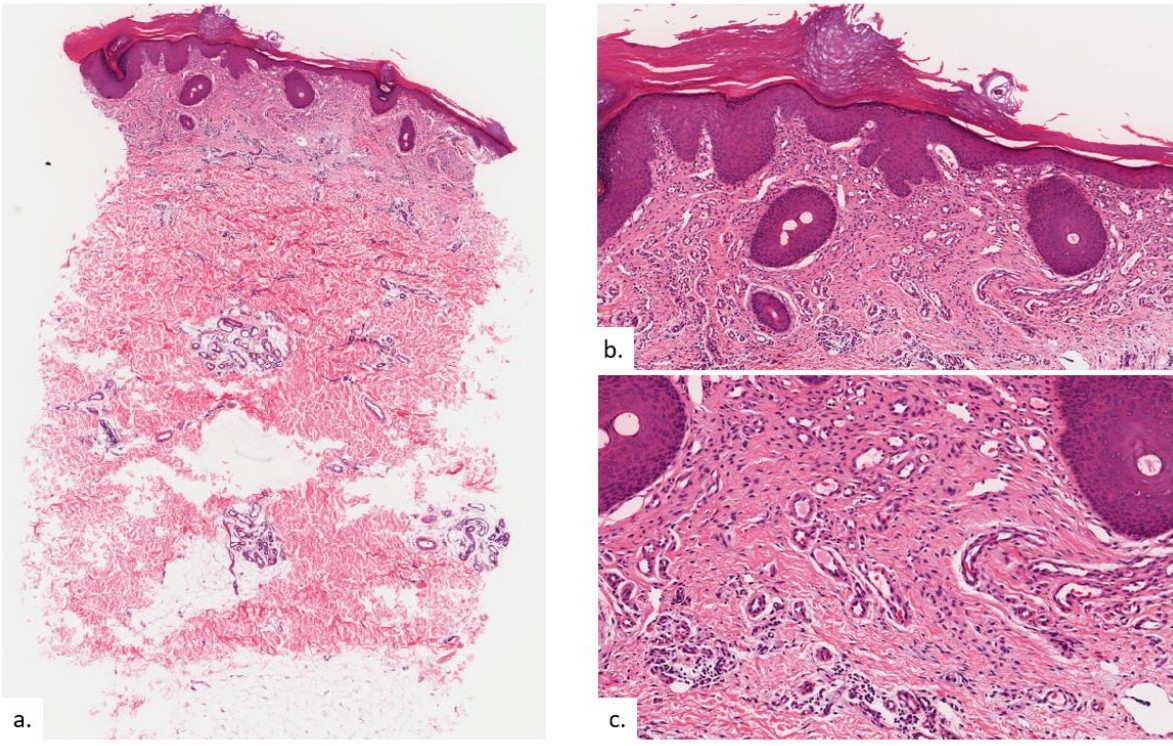

**Figure 6.** *Cont.*

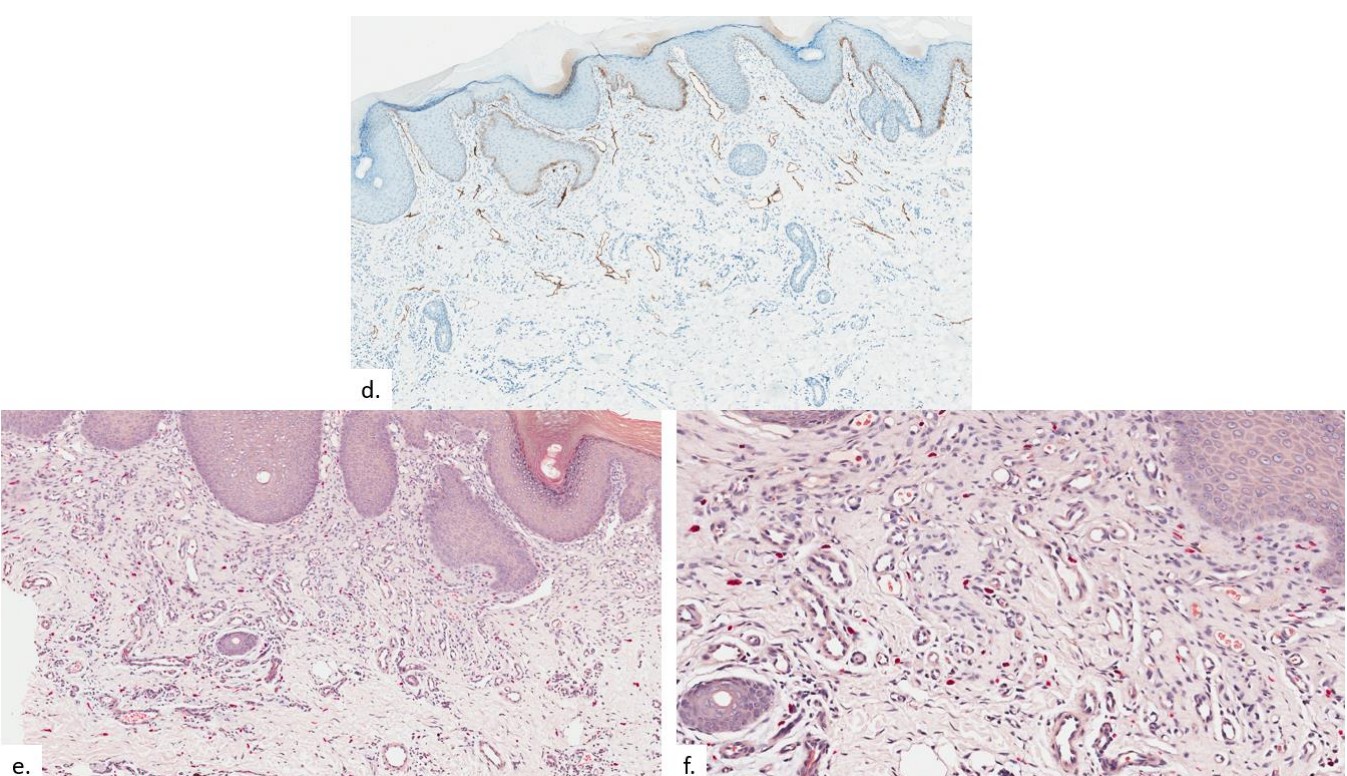

**Figure 6.** Patient 3. (**a**) There was compact hyperkeratosis and psoriasiform hyperplasia with hypergranulosis; (**b**) a vascular proliferation was present within the upper dermis; (**c**,**d**) the vascular proliferation consists of proliferation of capillary type vessels without endothelial cell atypia, with a modest increase in lymphatic channels on the podoplanin stain (**d**); (**e**,**f**) there were also increased quantities of MC on CAE.

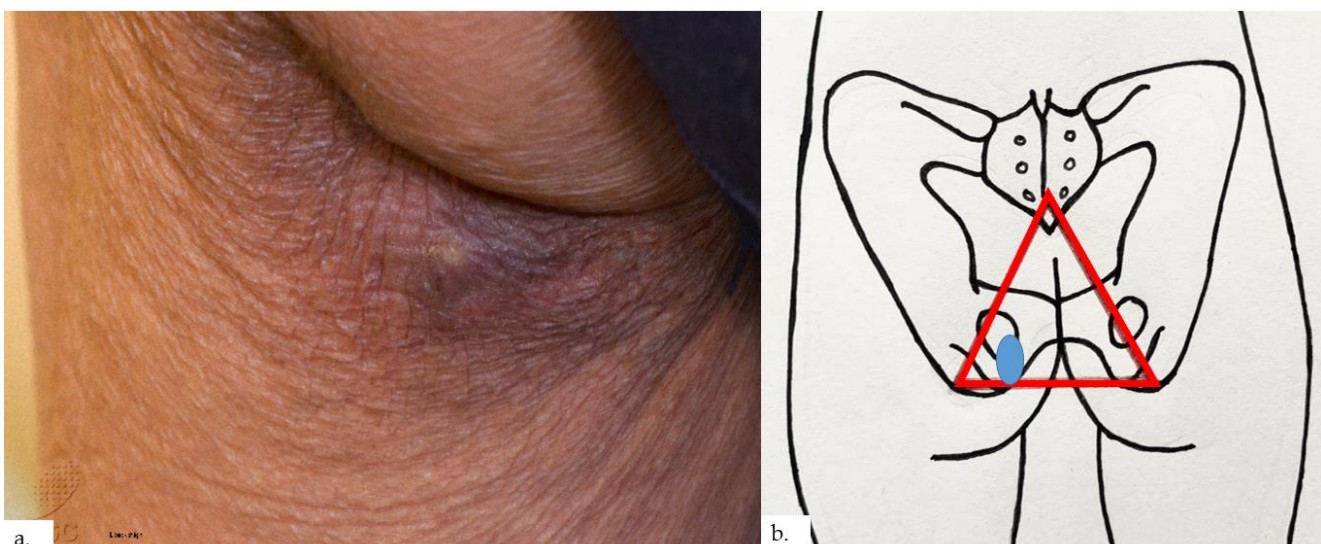

**Figure 7.** Patient 4. (**a**) There was a small tender 0.5 cm rounded dermal centered nodule over the left buttock. The surrounding skin showed hyperpigmentation and lichenification; (**b**) the nodule and surrounding lichenification were over the left gluteal fold, overlying the ischial tuberosity, at the base of the sitter's triangle (red triangle).

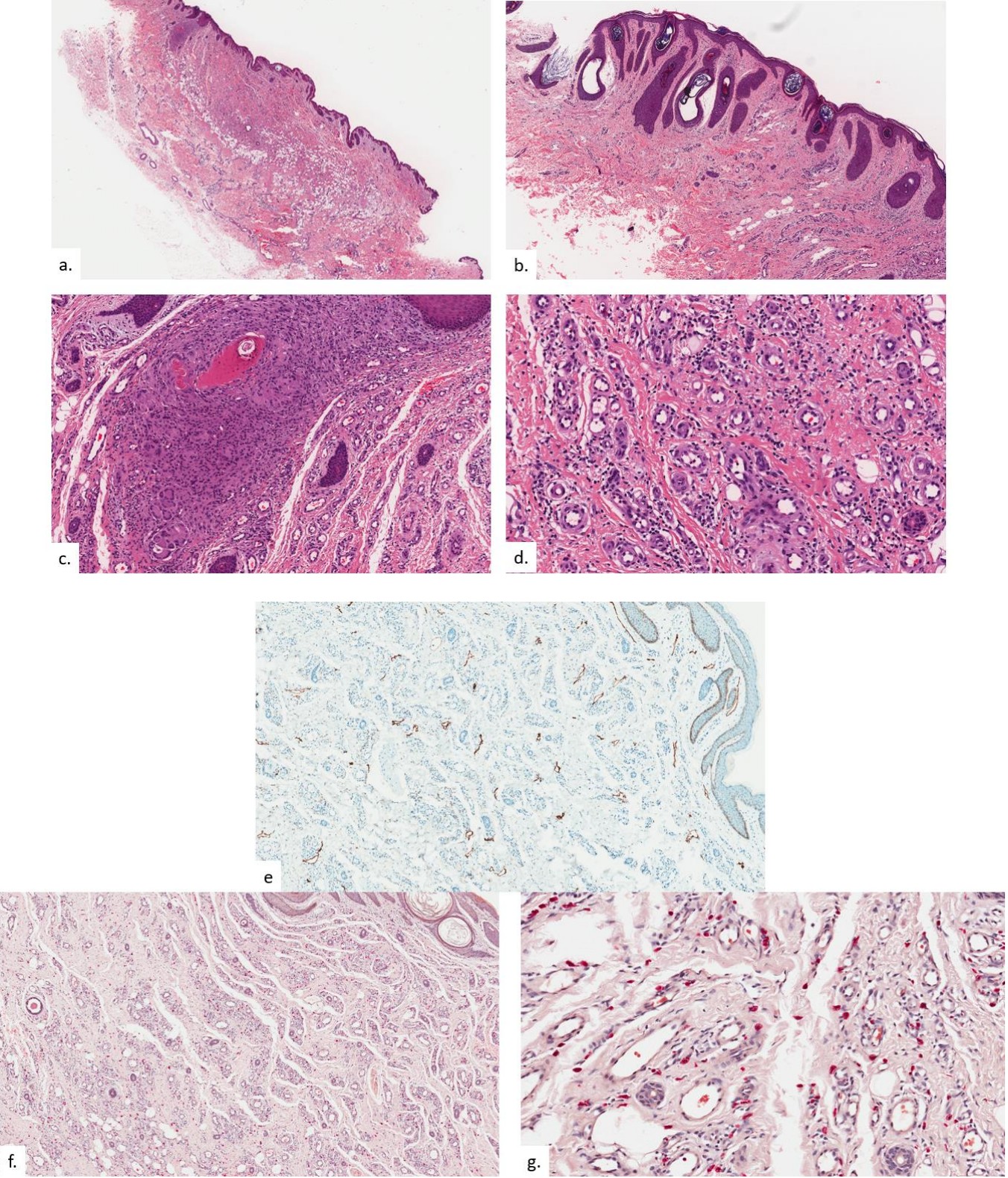

**Figure 8.** Patient 4. (**a**) There was compact hyperkeratosis, hyperplasia of the epidermis and follicular epithelium with a vascular proliferation throughout the dermis extending into the subcutis; (**b**) the epidermis was hyperplastic, with follicular plugging; (**c**) a focal foreign body granuloma was seen, with multinucleated giant cells surrounding keratinaceous debris; (**d,e**) there was a prominent pan-dermal proliferation of capillary-type vessels that extends into the subcutis, with a modest increase in lymphatic vessels noted on podoplanin staining; (**f,g**) increase in mast cell numbers is seen on CAE.

A summary of the clinical and histopathological features of these four patients is presented in Table 1.

**Table 1.** Characteristics and features of the four patients with REAR.

| | Case | | | |
|---|---|---|---|---|
| | **1** | **2** | **3** | **4** |
| Age (years) Sex Race | 68 Male Chinese | 73 Male Chinese | 68 Male Chinese | 94 Male Chinese |
| Co-morbidities | Hypertension, gastrointestinal reflux disease | Diabetes mellitus, hypertension, hyperlipidemia, asthma, benign prostatic hyperplasia, cognitive decline, pulmonary tuberculosis | Feet eczema | Hypothyroidism, atrial fibrillation with a pacemaker, benign prostatic hyperplasia |
| Location | Bilateral buttocks, inferio-lateral to the coccygeal apex | Bilateral buttocks, with a right lesion at the intergluteal fold, inferio-lateral to the coccygeal apex | Right intergluteal fold, inferio-lateral to the coccygeal apex | Left gluteal fold, overlying ischial tuberosity |
| Duration | 2 years | 2 years | 3 months | Few months |
| Symptoms | Intermittent pain in sitting | Mild discomfort on sitting | Mild discomfort on sitting | Asymptomatic but painful for a few weeks |
| Lesion Morphology | Symmetrical erythematous plaques, with some superficial scaling and surrounding mild lichenification | Slightly asymmetrical bilateral erythematous plaques, with scaling and lichenification. Blanchable erythema over the natal cleft | Unilateral erythematous plaque with scaling and surrounding erythema | A small tender 0.5 cm rounded dermal centered nodule, with hyperpigmentation and lichenification of the surrounding skin |
| Histological features | | | | |
| Hyperkeratosis | Y | Y | Y | Y |
| Psoriasiform hyperplasia | Y | Y | Y | Y |
| Extent of vascular proliferation | ++ | + | + | +++ |
| Location of vascular proliferation | Upper to mid dermis | Upper to mid dermis | Upper dermis | Dermis and subcutis |
| Lymphatic channels | + | ++ | + | + |
| Mast cell density | ++ | + | + | +++ |

(Y = Yes, N = No, + = mild amount, ++ = moderate amount, +++ = prominent amounts).

## 3. Discussion

Senile gluteal dermatosis (SGD) was first reported by Yamamoto et al. in 1979 as hyperkeratotic lichenified skin lesions of the gluteal region [1]. To date, most reports of this condition have arisen from Asia. There is a reported prevalence of 13% in elderly patients above 60 years of age, with a mean age of 70.4 years [2]. The increasing prevalence of SGD is seen with increasing age, lower lean body mass, and prolonged sitting [2–4]. Diabetes mellitus has also been associated with SGD [2]. The pathogenesis is related to prolonged pressure, repetitive friction, and hypoxia on the "sitter's triangle" formed by the coccygeal apex and bilateral ischial tuberosities [5,6], resulting in the development of hyperkeratotic, lichenified plaques on the gluteal area within the sitter's triangle, often flanking the perianal area. In our cases, 3 out of the 4 patients were found to have plaques over or near the intergluteal folds, inferiolateral to the coccygeal apex, whilst the plaque in the final patient

was over the ischial tuberosity. We postulate that plaques formed in the apex of the sitter's triangle, near the coccygeal apex, occur in patients with prolonged sitting positions at a reclined angle, putting more pressure on the coccygeal apex (Figure 9a). This reclined angle is commonly seen in modern office chairs and gaming chairs. In contrast, a prolonged upright 90° or inclined sitting angle places pressure on the ischial tuberosities; therefore, plaques formed would be located more at the base of the sitter's triangle (Figure 9b). The authors also postulate that microischaemia over bony pressure areas with prolonged sitting stimulates the characteristic reactive vascular proliferation seen within the dermis, which may sometimes be appreciated clinically as blancheable erythema of the skin, as in our second patient.

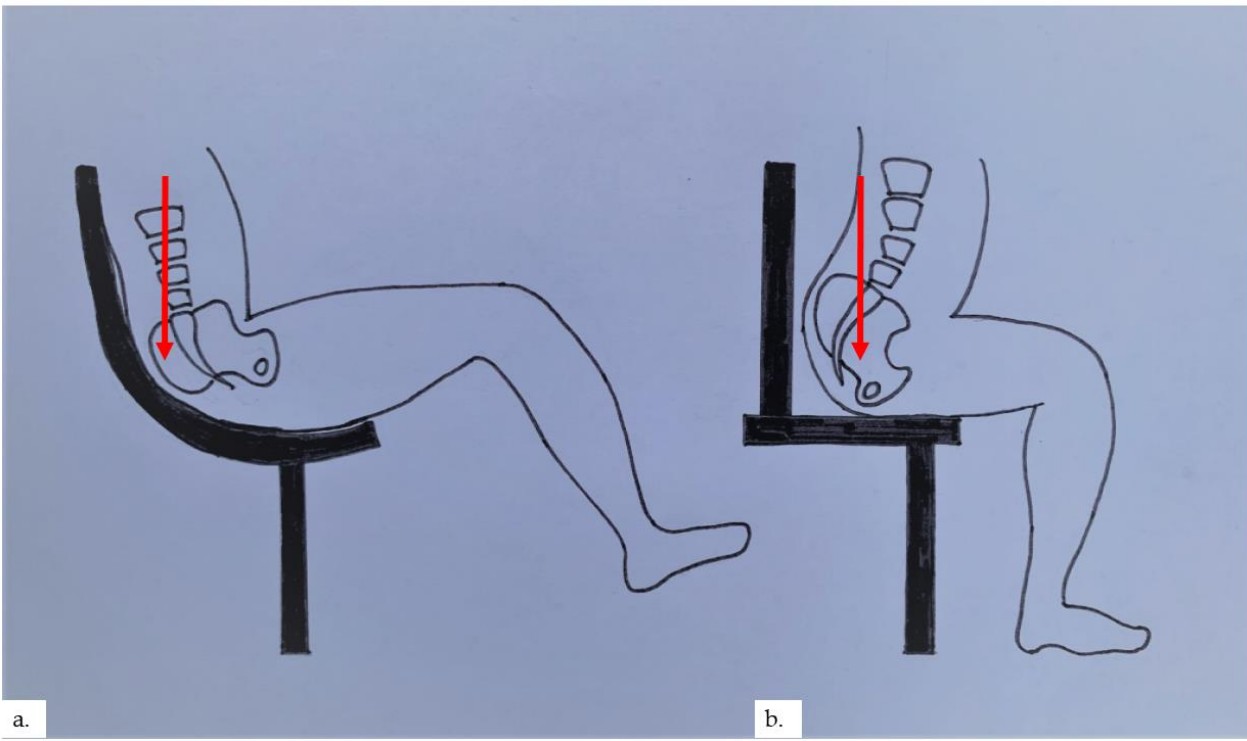

**Figure 9.** (**a**) Reclined sitting position places pressure on the coccygeal apex; (**b**) upright 90° sitting angle places pressure on the base of the sitter's triangle.

Clinical findings are the presence of unilateral or, more often, bilateral hyperpigmented and hyperkeratotic plaques in the gluteal region, often adjacent to the perianal area. The plaques are typically located overlying the bony prominences of the ischial tuberosities and the coccygeal apex, characteristically located within the sitter's triangle formed by these three bony prominences. The plaques are usually non-inflammatory, but there have been reports of psoriasiform-like plaques, erosions, and ulcerations [3,6]. Characteristic horizontal hyperkeratotic ridges may be seen in about half of the patients [3]. The plaques are usually asymptomatic, with a minority of the affected individuals describing itch or discomfort when sitting on hard surfaces [4]. This is in distinct contrast to clinical differential diagnoses such as prurigo which often has a persistent itch, and lichen simplex chronicus, which requires a history of repetitive scratching or picking. These findings are corroborated by our patients and summarized in Table 1.

The histopathological changes of SGD are usually mild and nonspecific [5]. Histological features include hyperkeratosis, psoriasiform epidermal hyperplasia and acanthosis, follicular plugging, and vascular dilatation/proliferation typically within the upper dermis with a sparse lymphohistiocytic perivascular infiltrate. More extensive changes reported have included ulceration with crust formation, papillary dermal edema, meandering vas-

cular proliferation extending into the reticular dermis, as well as a dense perivascular inflammatory infiltrate [2,4,5].

SGD also subsumes the recently suggested entity of prurigiform angiomatosis (PA), which is characterized by the clinical presentation of prurigo/lichen simplex chronicus-like epidermal hyperplasia with prominent dermal angioproliferation [7]. The authors found strong expression of VEGF in the upper two-thirds of the epidermis and VEGFR-2 expression in the endothelia of sprouting vessels. They postulate that chronic mechanical stress with subsequent relative hypoxia induces keratinocyte proliferation and expression of epidermal VEGF, resulting in neovascularization. In their study, all neovessels were CD31-positive and podoplanin-negative, consistent with capillaries and post-capillary venules. Interestingly, our study showed a modest increase in lymphatic channels amidst a proliferation of small blood vessels, as highlighted on podoplanin. Notably, in one case (patient 2), the proliferation of lymphatic vessels exceeded that of capillary-type vessels. This highlights that, at least in some cases, lymphatic-type vessels may contribute to neovascularization.

To our knowledge, our study is the first to describe mast cell (MC) infiltration in SGD. MC density appears to be positively correlated to the degree of vascular hyperplasia (patient 4 had a noteworthy degree of vasoproliferation, which correspondingly revealed the most intense MC staining on CAE). Marked MC infiltration has long been recognized in hemangiomas and vascular malformations; their roles have been more accurately delineated in recent years [8]. It has been elucidated that perivascular MC is pivotal in homeostatic vascular collateralization when triggered by intraluminal shear stress. Perivascular MCs are first activated by extravasated neutrophils via Nox2-derived reactive oxygen radicals, which in turn induce and promote both angiogenesis and lymphangiogenesis through the production of key mediators such as VEGF, TGFβ, TNF, and fibroblast growth factor-2 [9,10]. Additionally, mast cells may directly exert effects on vascular remodeling through increased MMP activity [11]. We postulate that the chronic decubitus pressure exerted on the "sitter's triangle" augments vascular shearing and, via the aforementioned mechanisms, engender reactive angio-/lymphangio-genesis to combat hypoperfusion. We also note that some patients with SGD experience itch. This can be explained as MC can secrete pruritogens via the classic IgE-mediated pathways (histamine and serotonin) or via the alternative but equally potent MRGPRX2 route (tryptase) [12]. This may explain why a subset of patients with SGD may scratch their lesions, resulting in features of lichenification histologically. For the majority of patients who do not experience itch, including all the patients in our series, the epidermal hyperplasia can largely be accounted for by repeated friction from recumbency.

While many patients may be elderly, SGD has been reported in younger patients who are in their 50s, making the term "senile" inappropriate. On the other hand, the term "prurigiform" in PA references the histological resemblance to prurigo/lichen simplex and also suggests that most cases are pruritic, when only a minority of patients (5–36%) experienced itch over their lesions [3,7]. To encompass the variable clinical findings encountered in this condition and concisely sum up the histopathological features seen, we propose a unifying name for Reactive Epidermal hyperplasia and Angiogenesis of the Rear (REAR).

Once recognized, the mainstay of treatment should be directed at pressure relieving measures. Other treatment options have shown varying degrees of success and include the use of topical steroids with or without keratolytic and topical tretinoin [2,3].

In summary, we present 4 cases of what has been described as senile gluteal dermatosis or prurigiform angiomatosis. We highlight novel findings of an increase in lymphatic vessels in addition to capillary-type neovascularization within lesions. We also note an increase in mast cell numbers in proportion to the degree of vascular proliferation. Lastly, we would like to propose a unifying terminology of Reactive Epidermal Hyperplasia with Angiogenesis of the Rear (REAR) to describe better and define this entity.

**Author Contributions:** Conceptualization, M.W.L. and J.S.S.L.; data writing—original draft preparation, M.W.L.; writing—review and editing, M.W.L., J.H.L.L., H.Y.C., S.-I.T. and J.S.S.L. All authors have read and agreed to the published version of the manuscript.

**Funding:** This research received no external funding.

**Institutional Review Board Statement:** The study was conducted in accordance with the Declaration of Helsinki and approved by the Institutional Review Board of the National Healthcare Group, Singapore, reference no. 2022/0074.

**Informed Consent Statement:** Informed consent was obtained from all subjects involved in the paper.

**Data Availability Statement:** Not applicable.

**Conflicts of Interest:** The authors declare no conflict of interest.

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
