# Peer review of "Reactive Epidermal Hyperplasia and Angiogenesis of the Rear (REAR): A Proposed Unifying Name for Senile Gluteal Dermatosis and Prurigiform Angiomatosis"

_dermatopathology, doi:10.3390/dermatopathology9040040_

Round 1

Reviewer 1 Report

The reviewer wishes to thank the editor and the authors for the opportunity to review this well written and extensively illustrated manuscript. The authors seek to further delineate the entity currently called senile gluteal dermatosis and re-name the entity, based on clinical and histopathologic data, Reactive Epidermal  hyperplasia and Angiogenesis of the Rear (REAR).  The manuscript is well illustrated and has a number of supportive diagrams that are much appreciated.

Overall, I like the new terminology and my only thought is that the manuscript may be more attractive if image resolution is increased in figures 2A, 4A, 4B, 4E, 6A, 8A, 8B, 8C, and 8D. Additionally, re-sizing of 8A to match the other images may be of help.

Also the key word should be changed from “Senile gluteal dermatitis” to “senile gluteal dermatosis” (line 17).

Author Response

Thank you for your review and comments! 

We will upload higher resolution pictures of the all the photomicrographs direct to the journal website and have re-sized 8A in the manuscript. 

We have also changed the key word “Senile gluteal dermatitis” to “senile gluteal dermatosis”. 

Reviewer 2 Report

The authors submit a well written article describing a small series of patients with so-called "senile gluteal dermatosis" or "prurigiform angiomatosis", proposing a unifying term REAR (reactive epidermal hyperplasia and angiomatosis of the rear). While the designation is certainly suggestive, I am afraid that it might be considered somewhat cheeky or insolent due to the use of informal language to describe a medical condition.

I also have a comment: the term "prurigiform" in "prurigiform angiomatosis", in my view, does refer to the pruritic nature of the lesions, but instead to the fact that the lesions histologically resemble nodular prurigo. In my opinion, this should be clarified in the manuscript.

Otherwise the cases presented are interesting, well documented, and typical of the entity described.

Author Response

Thank you for your review and comments! 

We wanted to propose an acronym that would emphasize on the location as well as highlight the characteristic histological features of the condition. In considering various options, we were mindful of the sensitivities pertaining to naming the gluteal region. However, the buttock location is central to recognizing this condition, and the acronym should adequately represent that. After much deliberation, we decided on REAR because it objectively described the location without sounding derogatory or offensive. 

We agree that the term prurigiform in "prurigiform angiomatosis" refers to its  resemblance histologically to prurigo/lichen simplex chronicus, and this is stated in the manuscript lines 187-189.

We will also include the reference to the histological resemblance to prurigo on page 12, highlighted in yellow:  "On the other hand, the term “prurigiform” in PA references the histological resemblance to prurigo/lichen simplex and also suggests that most cases are pruritic,"

Round 2

Reviewer 2 Report

I think the authors improved the manuscript and responded satisfactorily to the comments.